

# Multiple imputation methods: a case study of daily gold price

Ala Alrawajfi[1,2], Mohd Tahir Ismail[1], Sadam Al Wadi[3], Saleh Atiewi[4] and Ahmad Awajan[5]

[1] School of Mathematical Science, Universiti Sains Malaysia, Penang, Penang, Malaysia
[2] Department of Financial and Administrative Sciences, Ma'an College, Al-Balqa Applied University, Maan, Maan, Jordan
[3] College of Business, The University of Jordan, Amman, Amman, Jordan
[4] Department of Computer Science, Al-Hussein Bin Talal University, Maan, Maan, Jordan
[5] Department of Mathematics, Al-Hussein Bin Talal University, Maan, Maan, Jordan

## ABSTRACT

Data imputation strategies are necessary to address the prevalent difficulty of missing values in data observation and recording operations. This work utilizes diverse imputation methods to forecast and complete absent values inside a financial time-series dataset, specifically the daily prices of gold. The predictive accuracy of imputed data is assessed in comparison to the original entire dataset to ensure its robustness. The imputation methods are validated using actual closing price data obtained from a daily gold price website. The examined approaches include mean imputation, k-nearest neighbor (KNN), hot deck, random forest, support vector machine (SVM), and spline imputation. Their performance is evaluated based on several metrics, including mean error (ME), mean absolute error (MAE), root mean square error (RMSE), mean percentage error (MPE), and mean absolute percentage error (MAPE). The results indicate that the KNN approach consistently performs better than other methods in terms of all accuracy measures. Nevertheless, the precision of all techniques decreases as the proportion of missing data rises. Therefore, the KNN approach is suggested because to its exceptional performance and dependability in imputation tasks.

## INTRODUCTION

In today's financial environment, gold is an important indicator of economic stability and a safe investment option for investors. However, the accuracy of gold price research is sometimes compromised by the numerous missing values in databases. This study examines the challenges posed by incomplete data in current gold price analyses; the objective is to understand the effect on market knowledge and recommend novel approaches for addressing this crucial problem (*Jabeur, Mefteh-Wali & Viviani, 2024*).

The direction of gold prices affects the complex interaction among macroeconomic factors, market mood, and geopolitical concerns. Recently, global economic paradigms have experienced considerable, unprecedented changes, which have substantially influenced the behavior of investors and market dynamics. Thus, the intricacies of these changing

Corresponding authors
Ala Alrawajfi,
alaalrawajfi@student.usm.my
Mohd Tahir Ismail, m.tahir@usm.my

dynamics must be fully understood to make informed decisions and create effective risk management strategies in the financial sector (*Jabeur, Mefteh-Wali & Viviani, 2024*).

Various factors, such as data collection inconsistencies, reporting gaps, and technical constraints, are responsible for the missing values in gold price databases. These inconsistencies introduce extraneous data and partiality into investigations, thereby obstructing precise evaluations of market patterns and instability. Their consequences influence individual investors and could even affect financial stability on a large scale, posing systemic danger.

Multiple imputation (MI), a statistical approach, is frequently utilized to resolve missing data, and it provides practical solutions for handling incomplete data. MI procedures replace missing values with a set of credible values (instead of a single value) that reflect uncertainty about which value to substitute; the multiple imputed datasets are analyzed using standard techniques, and the analysis results are combined. The procedure of combining inferences from several datasets is generally similar regardless of the complete-data analysis performed (*Murray, 2018*).

Authors investigate techniques for imputation to estimate population means when there is nonresponse (*Chodjuntug & Lawson, 2022*). The authors concentrate on the application of these tools to analyze fine particle density data in Bangkok, specifically addressing the difficulties arising from missing data in environmental research. By conducting an analysis, they evaluate several imputation approaches to identify the most efficient approach for dealing with nonresponse. Their ultimate goal is to enhance the precision and dependability of population mean estimations in the context of air quality measurements (*Chodjuntug & Lawson, 2022*).

The authors in *Lawson (2023)* present a new imputation method in their study, which aims to estimate population means in the presence of missing data. This method utilizes a transformed variable approach. This approach is primarily utilized for analyzing air pollution data in Chiang Mai, Thailand, where the absence of data poses substantial obstacles to achieving precise analysis. The authors showcase the efficacy of their novel approach in enhancing the precision of population mean estimations, providing useful insights in the realm of environmental data analysis, and presenting a reliable strategy for handling partial datasets in air pollution research.

## RESEARCH OBJECTIVE

This study aims to relate theoretical developments in the analysis of current gold prices to practical applications. The objective is to evaluate the amount of missing data, identify market trends, and assess sophisticated imputation approaches to provide novel insights into the dynamics of gold prices and improve the reliability of market analysis. In short, This effort intends to equip financial data analysts with the necessary tools and information to conduct accurate and confident analysis of present financial markets..

This study utilizes the multiple imputation methodology, which is widely recognized as a powerful and flexible method for imputing missing data.. The initial step in this strategy is to generate several imputed datasets. Regression equations and various approaches, such

as KNN, spline, hot deck, support vector machine (SVM), and random forest, are utilized to predict missing data (*Liu et al., 2020*).

The article is structured as follows. 'Dataset' describes the dataset utilized in this research. The study's research approach is introduced in 'Research Methodology'. 'Missing Data Types and Mechanisms' shows the mathematical model and compares all imputation techniques. The assessment criteria are outlined in 'Mathematical Model', and the conclusions are given in 'Evaluation Criteria'.

## DATASET

Yahoo Finance provides gold price information that offers a complete perspective on the market performance of gold. This information includes data points, such as the highest, lowest, opening, and closing prices. These data offer an exhaustive overview of the volatility of gold prices on a daily, weekly, monthly, or yearly basis. Closing price denotes the ultimate valuation of gold during a particular trading session and indicates the final position of the market at the end of the day. This statistic is important for traders and investors because it accurately represents the mood and consensus reached by market players.

Meanwhile, the original valuation of gold at the start of a trading session is represented by the opening price. The opening price reveals the market sentiment before any substantial trading activity and serves as a benchmark that investors can utilize to evaluate the initial market dynamics. The maximum and minimum values achieved by gold during a certain period are represented by high and low prices, respectively; these prices denote the range within which gold was traded throughout the session. These measures are important for calculating the degree of volatility, identifying price patterns, and establishing specific trading strategies make good use of market moves.

Individuals, such as investors, analysts, and researchers, can utilize the gold price dataset from Yahoo Finance to perform a comprehensive analysis, establish patterns and correlations, and obtain essential market insights. This dataset assists in evaluating the effects of geopolitical concerns, macroeconomic events, and investor mood on gold prices and can therefore help individuals make well-informed investing decisions. Moreover, using historical data, users can test trading methods, evaluate risk exposure, and improve portfolio allocations to enhance their negotiation capability in the dynamic gold market. In this study, data on gold closing price obtained from Yahoo Finance in the period of 2000 to 2024 were analyzed.

The dataset in this research has undergone thorough analysis using R software to ensure its robustness. Table 1 presents the findings of the dataset analysis. The datasets with missing values were analyzed using a *p*-value test to determine if the missing values were missing completely at random (MCAR). Figure 1 shows that the *p*-value is greater than 0.05, indicating that our datasets with missing values are indeed MCAR. Additionally, a correlation test was conducted on the dataset, revealing a strong correlation of approximately 0.99 between the close price and all other variables.

| Table 1 | Gold close price dataset analysis. | | | | | |
|---------|------------|-------------|-----------|----------|-------------|-----------|
| **Dataset name** | **Data source** | **Time period** | **Number of records** | **Missing ratio** | **Number of observations** | **Number of variables** |
| Gold Price | Yahoo Finance | 2000–2024 | 5,828 | 10%–50% | 23,312 | 4 |

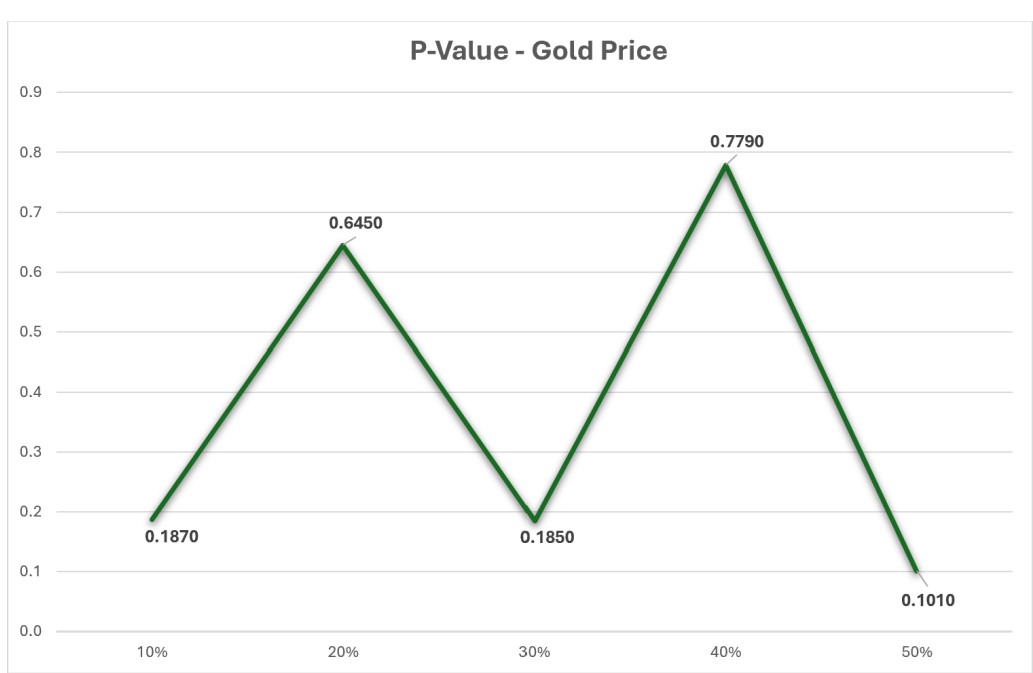

**Figure 1  P-value test for gold price dataset.**

# RESEARCH METHODOLOGY

This section can be further divided into two main subsections. The methods for making inferences about missing data are presented in the first subsection. The second subsection describes the assessment of the techniques' performance. This study applied a cross-validation approach on data covering the years 2000 to 2024 to evaluate the efficacy of various infilling strategies as shown in Table 2. Complete data for this time period were readily available, so these data were used as the baseline. The whole set of time-series data was subjected to a random simulation. Afterward, the data were examined to extract the missing daily streamflow data. Figure 2 shows the processes of preparing the dataset and inserting the missing data into the whole time series.

The simulation process for dataset imputation, which comprises four primary steps, is shown in Fig. 3. The first step was the collection of data on gold price from Yahoo Finance and the random generation of missing values in the collected dataset. The second step was the use of six different methods for data imputation. In the third step, the accuracies of the six methods were calculated using the original dataset and compared using the following criteria: mean error (ME), root-mean-squared error (RMSE), mean absolute error (MAE),

**Table 2    Sample data for the gold price (*Yahoo, 2024*).**

| Num | Date | Open | High | Low | Close |
|---|---|---|---|---|---|
| 1 | 29/08/2000 | 273.9 | 273.9 | 273.9 | 273.9 |
| 2 | 30/08/2000 | 274.8 | 278.3 | 274.8 | 278.3 |
| 3 | 31/08/2000 | 277 | 277 | 277 | 277 |
| 4 | 04/09/2000 | 275.8 | 275.8 | 275.8 | 275.8 |
| 5 | 05/09/2000 | 274.2 | 274.2 | 274.2 | 274.2 |
| 6 | 06/09/2000 | 274 | 274 | 274 | 274 |
| 7 | 07/09/2000 | 273.3 | 273.3 | 273.3 | 273.3 |
| 8 | 10/09/2000 | 273.1 | 273.1 | 273.1 | 273.1 |
| 9 | 11/09/2000 | 272.9 | 272.9 | 272.9 | 272.9 |
| …. | …. | …. | …. | …. | …. |
| …. | …. | …. | …. | …. | …. |
| 5,827 | 07/02/2024 | 2,032.8 | 2,039.5 | 2,032.7 | 2,035.2 |
| 5,828 | 06/02/2024 | 2,025.9 | 2,037.3 | 2,025.9 | 2,034.5 |

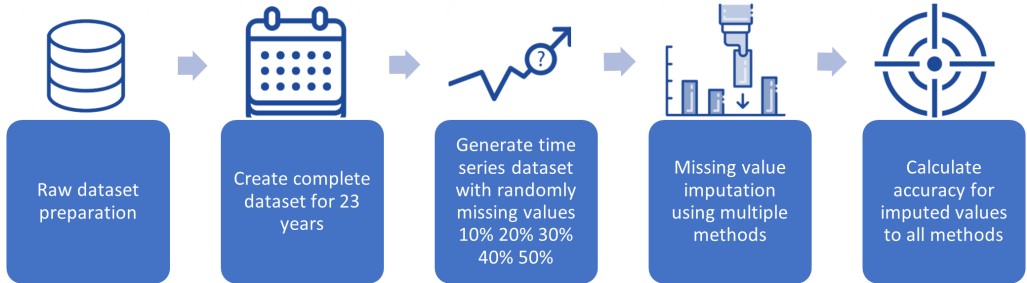

**Figure 2    Preparing the dataset and incorporating missing data into the entire time series.**

mean percentage error (MPE), and mean absolute percentage error (MAPE). In the fourth step, the results of the simulation were analyzed. The last three steps were performed 1,000 times each to achieve the highest possible level of precision.

## MISSING DATA TYPES AND MECHANISMS

The different forms of missing data can be classified using three categories. The link among the missing data mechanism, the seen values, and the missing values determines these categories. These categories need to be understood because the challenges presented by missing values and the approaches utilized to resolve these challenges differ for each of the three groups (*Pham, Pandis & White, 2022*).

Missing Completely at Random (MCAR). In MCAR, the missing values for an attribute do not rely on observable nor unobservable data. They are independent of both. In this type of randomness, bias of any sort cannot be added regardless of the treatment used for the missing data.

Missing at Random (MAR). In MAR, a missing value may be assumed to be reliant on any of the seen values, but it cannot be viewed as reliant on the unobserved data.

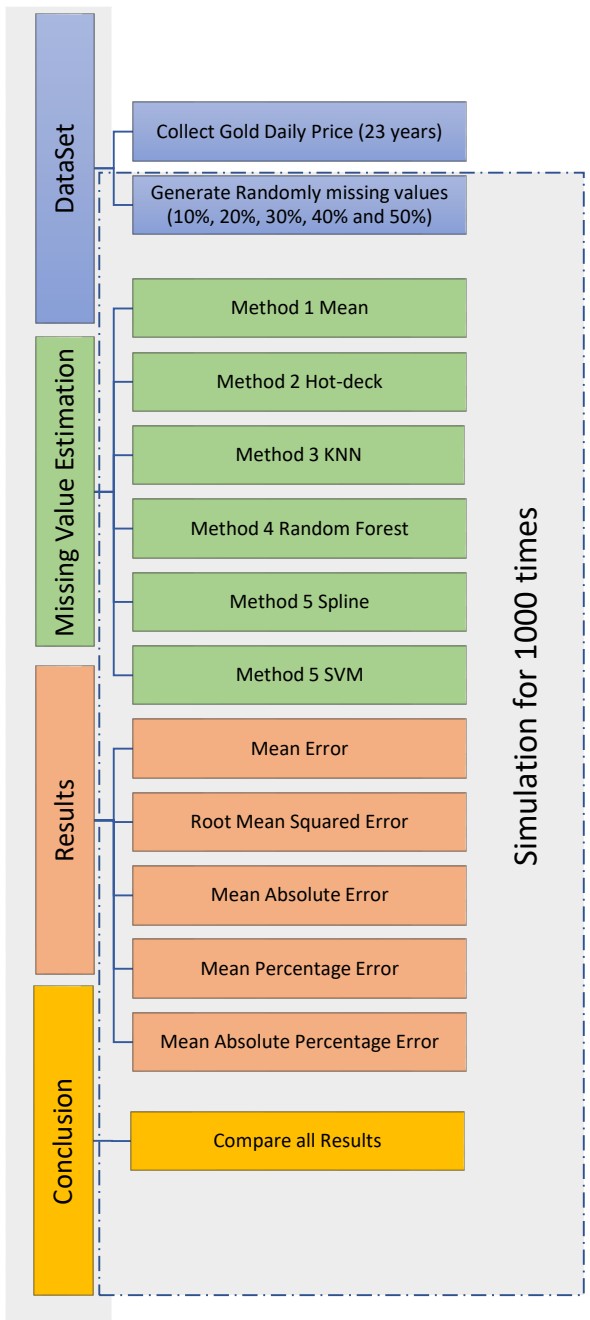

**Figure 3** Training and testing for dataset imputation.

Not Missing at Random (NMAR). In NMAR, a missing value is viewed as dependent on data that have not been seen. This kind of unpredictability can be addressed by a few approaches. Instance substitution is one of these techniques, and it may be used with mean or mode substitution. This naïve technique has the potential to induce bias, so it must be used with care. MCAR and MAR mechanisms are commonly regarded as ignorable, and non-ignorable missing value mechanisms are abbreviated as NMAR.

In this study, we applied the MCAR technique, which is commonly observed in the dataset. To introduce missing values, we utilized a specialized random function after retrieving the data from the original source. The missing values ranged from 10% to 50%.

# MATHEMATICAL MODEL

This study compared six imputation approaches, namely, mean, hot deck, KNN, spline imputation, random forest imputation, and SVM methods, to determine which among them is the most suitable for filling in the missing values in time-series datasets. First, a dataset from the time series data was arbitrarily removed from the primary dataset. The amounts of missing data ranged within 10%–50%, with 20%, 30%, 40%, and 50% being the most common. Several imputation approaches were employed to fill in all of the missing variables. The entire datasets produced by each method were compared with the dataset that was used initially to ascertain the accuracy of each approach. ME, MAE, RMSE, MPE, and MAPE were the accuracy parameters utilized to compare the five imputation methods.

The missing data in the gold price dataset were generated randomly at different rates ranging from 10% to 50%. The distribution of these missing values in the gold price dataset is shown in Fig. 4. The absent values are represented by red dots, and the values that are present are represented by blue points.

The organization of the dataset is given in Fig. 5. The dataset is divided into target data, feature data, and metadata. The target data denote the data that need to be imputed, and the feature data are used to help the imputation model find the missing values for the target data. The meta data describe and explain the data. The closing price was the target data in this study, and the imputation model's features were opening, high, and low prices. The meta data included number, date, and other information.

The imputation process is presented in Fig. 6. Data for the years 2000–2024 were collected from the Yahoo Finance website, saved in a CSV file, and imported into Orange data mining software (*Demšar et al., 2013*). Afterward, the features of the data were analyzed statistically by converting the data into a data table. After the data were transformed into a time series format, the metadata, features, and target value were identified. KNN was then utilized, and the complete dataset was saved in a data table format. Moreover, a line chart was created to visualize the imputed data, which were then compared with the original completed data to evaluate the imputation model's accuracy. The entire process was repeated for the other imputation methods.

Figure 6 depicts a data workflow in the Orange software that manages daily gold prices, including missing information, and does predictions. The following steps will be elucidated:
1.  Data loading (daily gold price with 10% missing value): This stage entails importing the dataset that includes daily gold prices, with 10% of the values being absent.

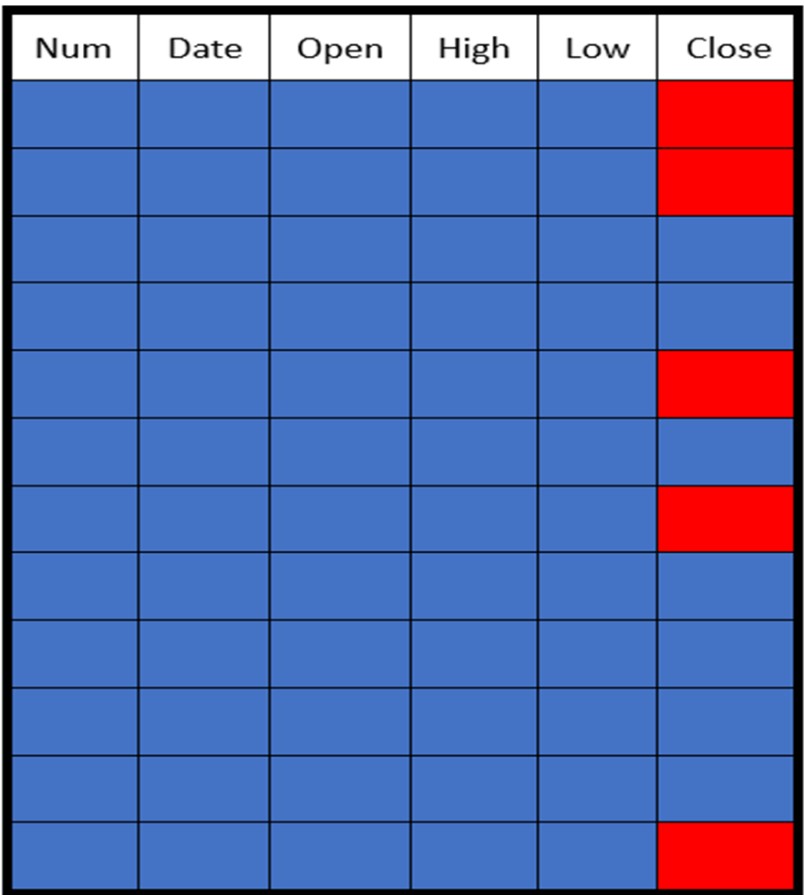

**Figure 4  Sample of a missing data pattern.**

2. Data table: The data that has been loaded is shown in a tabular format, which offers a comprehensive view of the dataset and enables the user to choose specific data points.
3. Feature statistics: This module computes and presents a range of statistical measures for the features inside the dataset. It aids in comprehending the distribution and attributes of the data.
4. Time series formation: This process is employed to transform the data into a format that follows a time series structure. It organizes the data in a suitable manner for doing time series analysis.
5. The KNN algorithm is a machine learning technique used for learning from time series data. It utilizes the values of the nearest neighbors to provide predictions for missing variables.
6. Evaluation and scoring: This step assesses the effectiveness of the KNN model by testing its performance and scoring its predictions. It aids in comprehending the precision and dependability of the model.
7. Impute: The dataset's missing values are filled in using the predictions made by the KNN model.

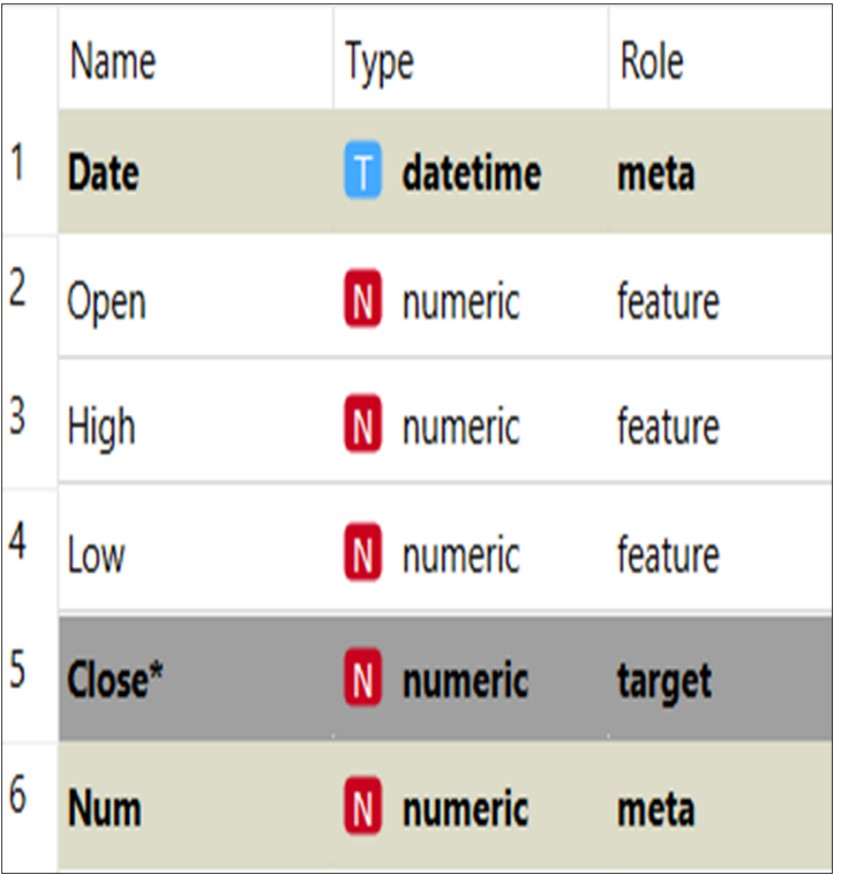

**Figure 5 Data set structure.**

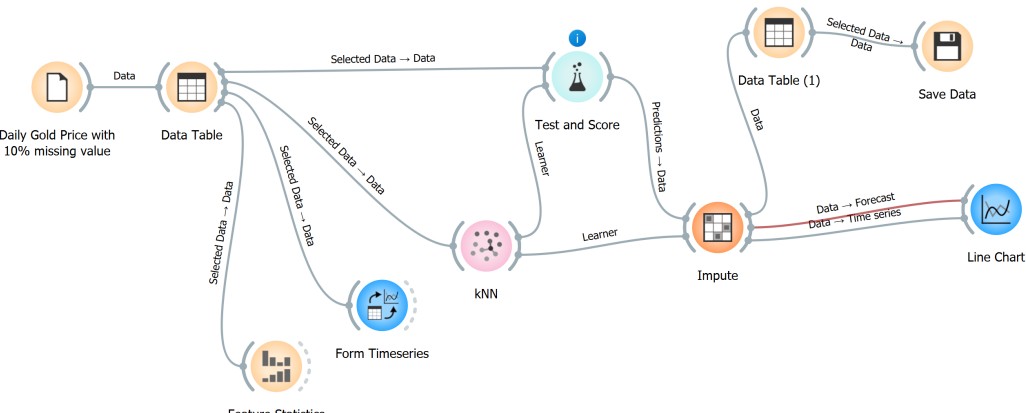

**Figure 6 Steps of data imputation using Orange software.** This Figure has been created using orange software https://orangedatamining.com/.

8. Data Table 2: Following the process of imputation, the data is once again presented in a tabular format to examine the filled values and verify that the dataset is now whole.

9. Store data: The finalized dataset is stored for future utilization or additional examination.

10. Line chart: The imputed data is represented graphically using a line chart, which facilitates the comparison between the actual and expected (imputed) values as they change over time. This visually confirms the accuracy of the imputation procedure.

Each stage in this workflow is interconnected, enabling a smooth and unified method to manage, manipulate, and examine time series data with missing values utilizing Orange software. The process was replicated for various models and varying missing ratios.

## Mean imputation

In mean imputation, a missing value for a particular variable is replaced by the average of currently available examples. This strategy is simple and does not change the sample size, but the estimates of standard deviation and variance are likely to be low because the strategy reduces the amount of variation in the data. Variation reduction also decreases the correlations and covariances, so this technique often produces incorrect estimations regardless of the underlying explanation for missing data (*Donders et al., 2006*).

The procedure and the mathematical framework for mean imputation are as follows:

1. Identify the missing value

   If we have a dataset with nobservations for a particular variable X, where $X_i$ represents the i−th observation. Some of these observations might be missing.

2. Calculate the mean of the observed values

3. Calculate the mean $\overline{X}$ of the non-missing values in the variable X using Eq. (1):

$$\overline{X} = \frac{1}{n-m} \sum_{i=1}^{n} X_i \tag{1}$$

   where $m$ is the number of missing values in $X$.

4. Impute the missing values

   Replace each missing value with the calculated mean $\overline{X}$ using Eq. (2):

$$X_{imputed,i} = \begin{cases} X_i & if\ X_i, is\ observed \\ \overline{X} & if\ X_i,\ is\ missing. \end{cases} \tag{2}$$

   Here, $X_{\text{imputed},i}$ represents the value of $X_i$ after imputation.

## Hot-deck imputation

Hot-deck imputation was extensively used in the past. In this approach, a missing value is substituted by the value of a record that was randomly selected and close to the value of the original record. The term "hot deck" was coined at the time when data were stored on punched cards. It means that the information donors are from the same dataset as the information receivers. The term hot stack of cards is used because it refers to being worked on just at that moment (*Christopher et al., 2019*).

Last observation carried forward (LOCF) is one of the methods of hot-deck imputation. First, LOCF produces an ordered dataset by sorting a dataset on the basis of any one of numerous factors. Second, the approach finds the first missing value and imputes it by

using the value of the cell that was present immediately before the missing value. The technique is applied repeatedly for each cell that contains a missing value until all of the missing values have been estimated. This technique resembles the opinion that if a measurement for a person or an entity is absent, the ideal assumption is that it has not changed from the previous time it was obtained. This situation is common in cases where the instances are repeated measurements of a variable for the same person or object. In other words, if a measurement is absent, the most reasonable assumption is that it has remained unchanged from the time it was last measured. However, the use of this strategy increases the likelihood of obtaining incorrect conclusions and contributes to an increase in bias (*Mavridis et al., 2019*; *Wongoutong, 2021*).

The procedure and the mathematical framework for hot-deck Imputation are as follows:

1. Define the dataset

   - Let $X = (x_{ij})$ be an $n \times p$ matrix of observed data, where n is the number of observations and p is the number of variables.
   - Let $M = (m_{ij})$ be an $n \times p$ matrix of missing data indicators, where $m_{ij} = 1$ if $x_{ij}$ is missing, and $m_{ij} = 0$ otherwise.

2. Identify donors and recipients

   - Define the set of donors $\mathfrak{D}$ and the set of recipients $\mathfrak{R}$:

   $$\mathfrak{D} = \{i | m_i j = 0 \setminus \text{ for all } j\}$$

   $$\mathfrak{R} = \{i | m_{ij} = 1 \setminus \text{ for some } j\}$$

3. Determine similarity
   - Define a similarity measure $S(i, k)$ between recipient i and donor k. Common choices include Euclidean distance, Mahalanobis distance, or other distance metrics.

4. Select the donor
   - For each missing value $x_{ij}$ (where $(i \in \mathfrak{R})$), select a donor $k$ (where $(k \in \mathfrak{D})$) such that $S(i, k)$ is minimized. This means finding the donor that is most similar to the recipient in terms of observed data.

5. Impute the missing value

   - Replace the missing value $x_{ij}$ with the corresponding value from the selected donor k:

   $$x_{ij} = x_{kj}.$$

## KNN imputation

KNN is a nonparametric supervised learning approach. It was introduced in 1951 by Evelyn Fix and Joseph Hodges for the field of statistics, and Thomas Cover developed it further. KNN is employed in regression and classification. The input is made up of $k$ training cases in a data collection that are the most similar to one another under both instances (regression and classification). Whether KNN is utilized for classification or regression decides the result (*Sanjar et al., 2020*).

Class membership is the outcome of KNN classification. How the object should be categorized is voted by a plurality of an item's neighbors. Then, the object is placed in the class that has the highest frequency among the item's $k$ closest neighbors ($k$ is a positive integer and often small). The item in question is viewed as belonging to the category of its one closest neighbors when $k$ is equal to 1. KNN regression generates the value of a certain attribute associated with an object, and this value is computed by obtaining the average of the values of the $k$ closest neighbors (*Sanjar et al., 2020*). In KNN classification, the function is only approximated locally, and the entire computation is postponed until the function is assessed. If the features represent different physical units or have considerably different scales, then training data normalization can remarkably improve the accuracy of the algorithm because the classification of this algorithm is based on distance. Notably, the distance between features can exhibit considerable variation (*Bania & Halder, 2020*).

The assignment of weights to the contributions provided by neighbors may be a good classification and regression strategy. It ensures that the influence of neighbors situated close to the center of the map on the general average is greater than the influence of neighbors located far from the map's center. For example, in a common technique of weight calculation, each neighbor is assigned the number 1/d as its weight; d is the distance that separates the subject and the neighbor (*Jadhav, Pramod & Ramanathan, 2019*).

In KNN classification or KNN regression, the neighbors are selected from a set of objects in which the class (for KNN classification) or the object (for KNN regression) attribute value has already been determined. Although an explicit training phase is unnecessary, this set may still be regarded as the algorithm's training set (*Jadhav, Pramod & Ramanathan, 2019*).

1. Identify the missing values:

   - Let $X$ be the dataset with nsamples and features. Identify the missing value(s) in $X$. Suppose $x_{i,j}$ is the missing value in the $i-$th sample and $j-$th feature.

2. Calculate distances
   - For each sample i with a missing value, calculate the distance to all other samples that have a known value in the $j-th$ feature. A common distance metric used is the Euclidean distance. For two samples pandq, the Euclidean distance is given by Eq. (3):

$$d\left(p,q\right) = \sqrt{\sum_{k=1}^{m}\left(x_{p,k} - x_{q,k}\right)^2} \tag{3}$$

   where the summation is over all features kfor which $x_{p,k}$ and $x_{q,k}$ are not missing.

3. Select nearest neighbors

   - Choose the K nearest neighbors to the sample i based on the calculated distances. Let these neighbors be indexed by N(i).

4. Impute the missing value
   - There are different strategies to impute the missing value using the nearest neighbors:
   o   Mean imputation using Eq. (4)

$$x_{i,j} = \frac{1}{K}\sum_{q\in N(i)} x_{q,j} \tag{4}$$

o  Median imputation using Eq. (5)

$$x_{i,j} = \text{median} x_{q,j} : q \in N(i) \tag{5}$$

## Random forest imputation

One of the ensemble learning techniques is random forest. Throughout the training phase, this technique creates many decision trees, and it consolidates the forecasts generated by separate decision trees. The random forest method's initial stage entails the construction of decision trees. A random subset of the training data is used to build each decision tree inside the random forest. A randomly selected subset of characteristics is evaluated for splitting at every node of the decision tree. Such randomization reduces the correlation between individual trees and mitigates the overfitting problem. In the process of training, each decision tree is constructed by randomly selecting (with replacement) a portion of the training data. A subset of characteristics for every node in a decision tree is then randomly chosen. The selected characteristics are utilized to identify the optimal division at that particular node (*e.g.*, the use of Gini impurity for classification or mean squared error for regression). The procedure is continuously repeated until the tree has reached its maximum size (or until a condition for stopping has been met). In the process of prediction, the input characteristics are made to pass through each decision tree in the forest (the third phase). In classification tasks, each tree votes for a certain class, and the class with the most votes is chosen as the ultimate prediction. In regression tasks, each tree continuously generates value predictions, and the final prediction is obtained by taking the average or median of these forecasts. Aggregation is the final step in the random forest process. During aggregation, the aggregated forecast is acquired by combining the predictions from all the individual trees. In classification tasks, this process may involve determining the mode that represents the most common prediction. In regression analysis, the average or median value of the predicted outcomes is calculated (*Ren et al., 2023*).

The process of using random forest for imputation involves the following steps:

1. **Initial imputation:**
   Fill the missing values with initial guesses, often using mean/mode for numerical/categorical variables.

2. **Iterative imputation:**
   b) For each feature $j$ with missing values, separate the dataset into two parts: $D_{obs}$ (where $j$ is observed) and $D_{miss}$ (where $j$ is missing).
   a) Train a random forest model $\mathfrak{F}_j$ on $D_{obs}$ using all other features to predict $j$.
   c) Predict the missing values in $D_{miss}$ using $\mathfrak{F}_j$.
   d) Replace the initial guesses of the missing values in $\mathfrak{F}_j$ with the predictions from $\mathfrak{F}_j$.

3. **Repeat**
   Iterate over all features with missing values, updating the imputations until the imputations converge or a set number of iterations is reached
   - Mathematical formulation
   Given a dataset $X$ with n samples and $p$ features, let $x_{ij}$ denote the value of feature $j$ for sample $i$. Let $X_{\neg j}$ represent all features except $j$.

1. **Initial imputation:**
   $\hat{X}_{ij}^{(0)} = $ initial guess for missing value.
2. **Iterative imputation:** For iteration $t$:
   (a) For each feature j:
      i. Train Random Forest $F_{j(t)on}\{(x_{i,\neg j}, x_{ij}) \,|\, x_{ij \text{ i osre}}\}$.
      ii. Predict missing values using Eq. (6):

$$\hat{x}_{ij}^{(t)} = F_{j^{(t)}}(x_{i,\neg j}). \tag{6}$$

   (a) Update the imputed values using Eq. (7):

$$x_{ij}^{(t+1)} = \hat{x}_{ij}^{(t)}. \tag{7}$$

3. **Convergence:** Continue until convergence or maximum iterations.

## Spline imputation

The interpolant in spline imputation is a particular kind of piecewise polynomial known as a spline. Spline imputation is a type of interpolation. Spline interpolation fits nine cubic polynomials between each pair of 10 points instead of fitting a single 10° polynomial to all the points. This approach is different from the traditional high-degree linear interpolation method that fits a single high-degree polynomial to all the points simultaneously. Spline interpolation is often preferred over polynomial interpolation because the former's interpolation error is small even when low-degree polynomials are used for the spline. Spline interpolation is employed in various applications. Runge's phenomenon, in which oscillation occurs between points during interpolation by high-degree polynomials, is prevented by spline interpolation (*Tayebi, Momani & Arqub, 2022*).

The process of using Spline imputation involves the following steps:

Given a set of observed data points $(x_1, y_1), (x_2, y_2), \ldots, (x_n, y_n)$ with some missing values, spline interpolation aims to fit a smooth curve $f(x)$ as in Eq. (8):

$$f(x_i) = y_i \quad \text{for all observed } (x_i, y_i). \tag{8}$$

The most common spline used is the cubic spline, which is a piecewise cubic polynomial. The cubic spline $S(x)$ is defined in Eq. (9):

$$S(x) = \begin{cases} a_1 + b_1(x - x_1) + c_1(x - x_1)^2 + d_1(x - x_1)^3 & \text{for } x \in [x_1, x_2] \\ \vdots & \vdots \\ a_{n-1} + b_{n-1}(x - x_{n-1}) + c_{n-1}(x - x_{n-1})^2 + d_{n-1}(x - x_{n-1})^3 & \text{for } x \in [x_{n-1}, x_n] \end{cases} \tag{9}$$

● Conditions for cubic splines

**Continuity:**      The spline $S(x)$ must be continuous at each data point $x_i$.

$$S(x_i) = y_i \quad \text{for all } i$$

**Smoothness:**      The first and second derivatives of the spline must be continuous at each internal data point.

$$S'(x_i^-) = S'(x_i^+) \quad \text{and} \quad S''(x_i^-) = S''(x_i^+) \quad \text{for all internal } x_i.$$

• Boundary conditions: There are several choices for boundary conditions, but common ones include:

**Natural spline:**      The second derivative at the endpoints is zero.

$$S^{''}(x_1) = 0 \quad \text{and} \quad S^{''}(x_n) = 0.$$

**Clamped spline:**      The first derivative at the endpoints is specified.

$$S^{'}(x_1) = f^{'}(x_1) \quad \text{and} \quad S^{'}(x_n) = f^{'}(x_n).$$

Solving for coefficients

To find the coefficients $(a_i, b_i, c_i, d_i)$ for each segment, we set up a system of linear equations based on the conditions above and solve for the unknowns.

## SVM imputation

SVM is widely recognized in data imputation because it can properly address high-dimensional data and understand intricate connections within the data. SVM identifies the advantageous hyperplane that divides distinct classes or data points in a space with several dimensions. SVM may be utilized to predict missing values in data imputation because it can identify patterns and connections from available data (*Dudzik, Nalepa & Kawulok, 2024*).

The SVM process involves different stages. At the **feature extraction stage**, the dataset is expressed as a set of distinct characteristics or variables. These attributes characterize each observation or data point. At the **training stage**, the SVM algorithm learns from the entire observed data and determines the most advantageous among hyperplanes to effectively distinguish between several classes or data points in the feature space. At the **imputation stage**, the SVM model is trained and subsequently used to predict missing values in the dataset on the basis of the associations learned at the training stage. The values of other characteristics in the dataset are utilized by the model to forecast the missing values. At the **evaluation stage**, the correctness and consistency of the imputed values are assessed. The quality of the imputation is evaluated using various metrics and methodologies, including MSE or cross validation. At the **integration stage**, the gaps in the data are filled by incorporating the inferred values into the dataset (*Dudzik, Nalepa & Kawulok, 2024*).

The process of using SVM imputation involves the following steps:

1. Problem definition given a dataset $X = \{(x_1, y_1), (x_2, y_2), \ldots, (x_n, y_n)\}$ with missing values, the goal is to predict the missing values $y_i$ for the corresponding $x_i$.

2. Data preparation

   • Feature matrix: $X$ is the feature matrix where each row represents an instance and each column represents a feature
   • Target Vector: $y$ is the target vector containing the values to be predicted. Some entries in $y$ are missing.

3. Handling missing data

- Separate the data into two parts: observed data $(X_{\text{obs}}, y_{\text{obs}})$ and missing data $(X_{\text{miss}}, y_{\text{miss}})$.
- $X_{\text{obs}}$ and $y_{\text{obs}}$ represent the complete cases, while $X_{\text{miss}}$ and $y_{\text{miss}}$ represent the incomplete cases where $y_{\text{miss}}$ needs to be predicted.

4. Training the SVM model

   - Objective function: The SVM aims to find a hyperplane that best separates the data into classes (for classification) or fits the data (for regression).
   - Support vectors: These are the data points that are closest to the hyperplane and determine its position.
   - Kernel function: $K(x_i, x_j)$ transforms the input data into a higher-dimensional space to make it easier to find a separating hyperplane.
   - Optimization problem: (for regression) using Eq. (10):

$$\min_{w,b,\xi,\xi^*} \frac{1}{2}|w|^2 + C\sum_{i=1}^{n}(\xi_i + \xi_i^*) \tag{10}$$

   subject to Eqs. (11) and (12)

$$y_i - (w \cdot \phi(x_i) + b) \leq \epsilon + \xi_i \tag{11}$$

$$(w \cdot \phi(x_i) + b) - y_i \leq \epsilon + \xi_i^* \tag{12}$$

$\xi_i, \xi_i^* \geq 0$

5. Predicting missing values

   Use the trained SVM model to predict the missing values $y_{\text{miss}}$ for the incomplete cases $X_{\text{miss}}$ using Eq. (13):

   $$\widehat{y_i} = w \cdot \phi(x_i) + b. \tag{13}$$

   The predicted values $(\widehat{y_i})$ are the imputed values for the missing entries in $y_{\text{miss}}$.

## EVALUATION CRITERIA

This research compared imputation methods by using five performance measures.

### ME

The average of all the individual mistakes in a collection is called ME. Measurement uncertainty or the disparity between measured and real values is referred to as an error in this context. Error is formally called measurement or observational error (*Karunasingha, 2022*). Equation (14) can be used to derive ME.

$$ME = \frac{\sum_{i=1}^{n} y_i - x_i}{n}. \tag{14}$$

## MAE

The differences in the results derived from two observations that exhibit similar phenomena is measured by MAE. Examples of $Y$ versus $X$ include comparisons of initial time with future time, one measurement method *versus* another measurement technique, and one measurement technique *versus* an alternate measurement technique. MAE is calculated as the total number of absolute errors divided by the number of samples (*Ji & Peters, 2003*). Equation (15) shows the derivation of MAE.

$$MAE = \frac{\sum_{i=1}^{n} |y_i - x_i|}{n}.$$  (15)

## RMSE

RMSE refers to the difference between actual and expected values given as a standard deviation value (prediction error). The residuals measure how far the data points deviate from the regression line, and RMSE shows how distributed these residuals are. Both measurements are referred to as RMSE. RMSE provides indicates the extent to which the data are centered around the best-fit line. RMSE is a common method to confirm the accuracy of experimental data in the fields of climatology, forecasting, and regression analysis (*Karunasingha, 2022*). RMSE can be expressed as in Eq. (16).

$$RMSE = \sqrt{\frac{\sum_{i=1}^{n} (y_i - x_i)^2}{n}}.$$  (16)

## MPE

MPE is the average of percentage errors. It determines the extent of the difference between forecasts of a model and the actual values of the quantity being forecasted (*Karunasingha, 2022*). Equation (17) shows the derivation of MPE.

$$MPE = \frac{100\%}{n} \sum_{i=1}^{n} \frac{y_i - x_i}{y_i}$$  (17)

## MAPE

In statistics, MAPE measures the prediction accuracy of a forecasting approach. Accuracy is often expressed as a ratio in MAPE (*Awajan, Ismail & AL Wadi, 2018*; *Karunasingha, 2022*). Equation (18) shows how MAPE is obtained.

$$MAPE = \frac{100\%}{n} \sum_{i=1}^{n} \left| \frac{y_i - x_i}{y_i} \right|$$  (18)

**Table 3  Accuracy results for all imputation methods.**

|  | Method | 10% | 20% | 30% | 40% | 50% |
|---|---|---|---|---|---|---|
| **ME** | Mean | 2.622657 | 2.81494 | 3.69881 | 2.397894 | 5.678135 |
|  | Hot-deck | 0.0068 | 0.025359 | 0.140231 | 0.20021 | 0.236354 |
|  | KNN | 0.036389 | 0.004258 | 0.102867 | 0.044145 | 0.031474 |
|  | Random forest | 0.010231 | 0.043482 | 0.090737 | 0.075594 | 0.126948 |
|  | Spline | 0.000915 | 0.008411 | 0.054286 | 0.059011 | 0.365068 |
|  | SVM | 1.284972 | 3.425251 | 3.773801 | 4.365908 | 5.903704 |
| **MAE** | Mean | 43.8372 | 90.83837 | 139.0055 | 183.169 | 227.5104 |
|  | Hot-deck | 0.418777 | 0.908745 | 1.298712 | 1.886675 | 2.402195 |
|  | KNN | 0.352211 | 0.689023 | 1.075983 | 1.490204 | 1.899488 |
|  | Random forest | 0.346103 | 0.725953 | 1.096472 | 1.562646 | 2.009779 |
|  | Spline | 0.641019 | 1.368196 | 2.195195 | 3.142399 | 4.065315 |
|  | SVM | 39.26009 | 81.79856 | 125.1141 | 166.0619 | 204.7144 |
| **RMSE** | Mean | 163.0496 | 238.6665 | 296.4559 | 339.6391 | 376.7184 |
|  | Hot-deck | 1.973288 | 3.273413 | 3.624834 | 4.883218 | 5.253412 |
|  | KNN | 1.66818 | 2.43726 | 3.146221 | 3.682833 | 4.209081 |
|  | Random forest | 1.677957 | 2.610471 | 3.203094 | 4.067637 | 4.558986 |
|  | Spline | 2.974924 | 4.54611 | 6.117541 | 7.331447 | 8.837399 |
|  | SVM | 146.4295 | 214.8564 | 266.4293 | 306.6426 | 338.0913 |
| **MPE** | Mean | 3.741063 | 9.568722 | 14.85041 | 17.78845 | 21.59195 |
|  | Hot-deck | 8.86E−06 | 0.001443 | 0.013207 | 0.01997 | 0.028669 |
|  | KNN | 0.001649 | 0.001063 | 0.008596 | 0.003968 | 0.003494 |
|  | Random forest | 0.000161 | 0.005276 | 0.008306 | 0.006202 | 0.008917 |
|  | Spline | 0.000479 | 0.004695 | 0.001065 | 0.006857 | 0.019815 |
|  | SVM | 3.498386 | 8.737487 | 13.4188 | 15.78457 | 19.29079 |
| **MAPE** | Mean | 6.583765 | 14.91244 | 23.04417 | 28.96311 | 35.70273 |
|  | Hot-deck | 0.038403 | 0.08616 | 0.123787 | 0.175933 | 0.226499 |
|  | KNN | 0.032525 | 0.065792 | 0.100593 | 0.137439 | 0.176536 |
|  | Random forest | 0.032335 | 0.068432 | 0.103105 | 0.143321 | 0.185198 |
|  | Spline | 0.056902 | 0.125945 | 0.201119 | 0.283404 | 0.3663 |
|  | SVM | 5.971745 | 13.4933 | 20.77255 | 26.08142 | 32.05835 |

# DISCUSSION AND RESULTS

Table 3 shows accuracy results for all imputation methods, Figs. 7–11 do not show mean and SVM imputation methods because the results indicate that these approaches had the lowest accuracy among all the imputation methods.

The MEs of 5,882 records of daily gold price are given in Fig. 7. KNN, random forest, hot deck, and spline imputation methods were compared, and the results revealed that when the missing data percentages were 40% and 50%, KNN had the smallest ME among all the methods; it was followed by random forest. The spline method performed well when the missing data percentage was 10%–30%. The results also showed that ME increased when the missing data percentage increased.

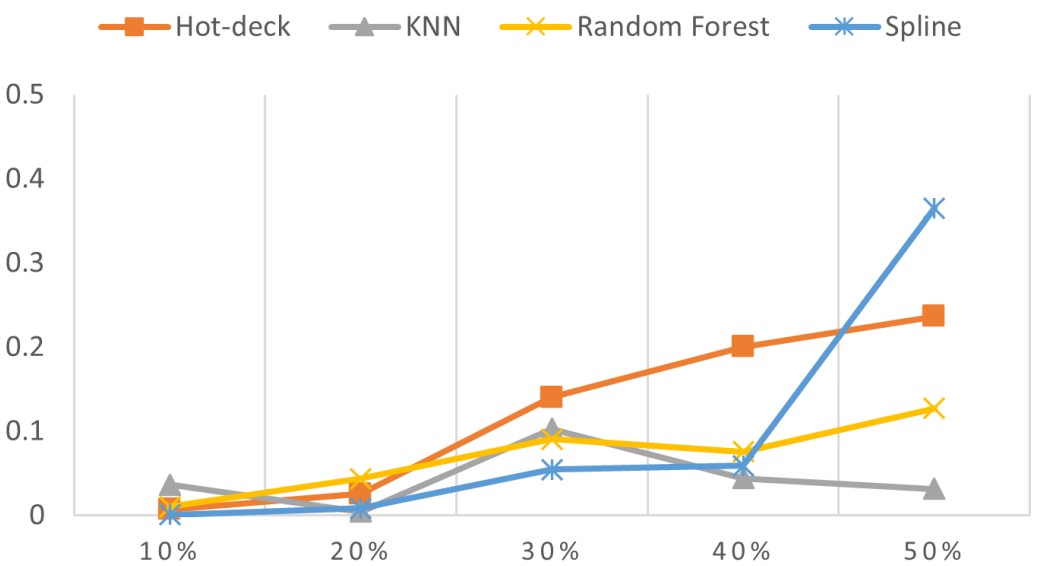

**Figure 7** **MEs of 5,882 records of daily gold price.**

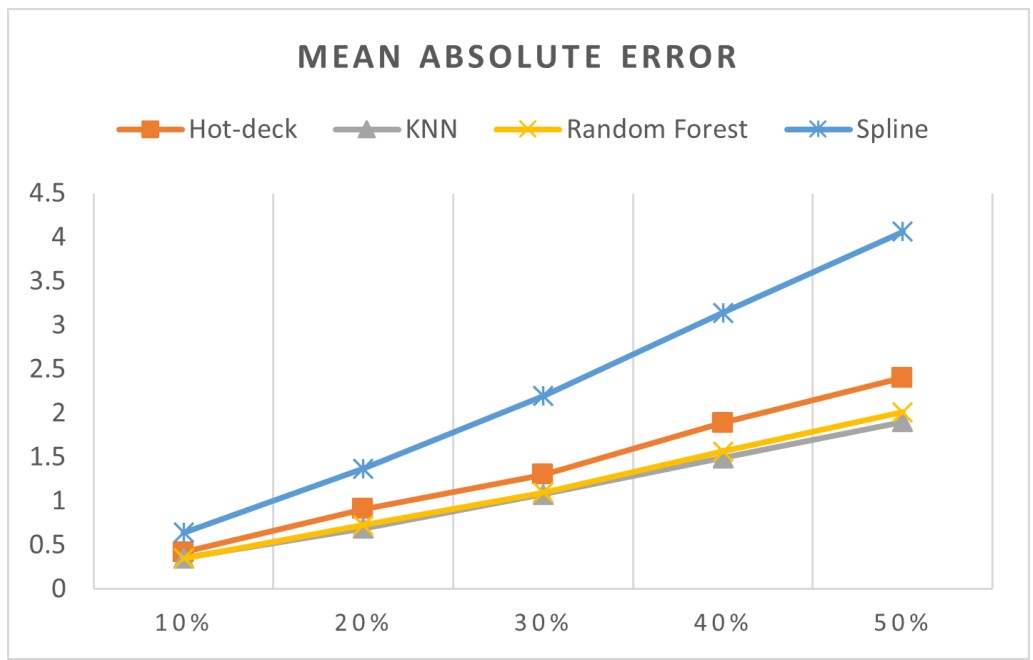

**Figure 8** **MAEs of 5,882 records of daily gold price.**

ROOT MEAN SQUARED ERROR

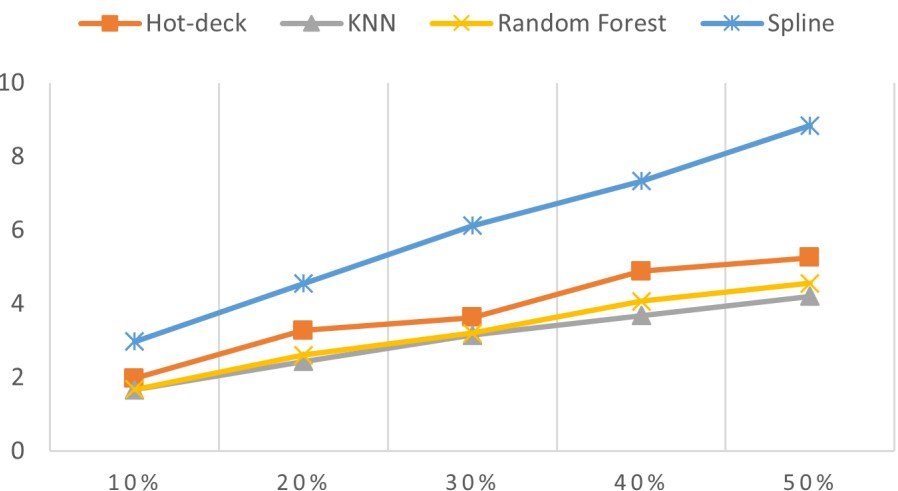

**Figure 9** RMSEs of 5,882 records of daily gold price.

MEAN PERCENTAGE ERROR

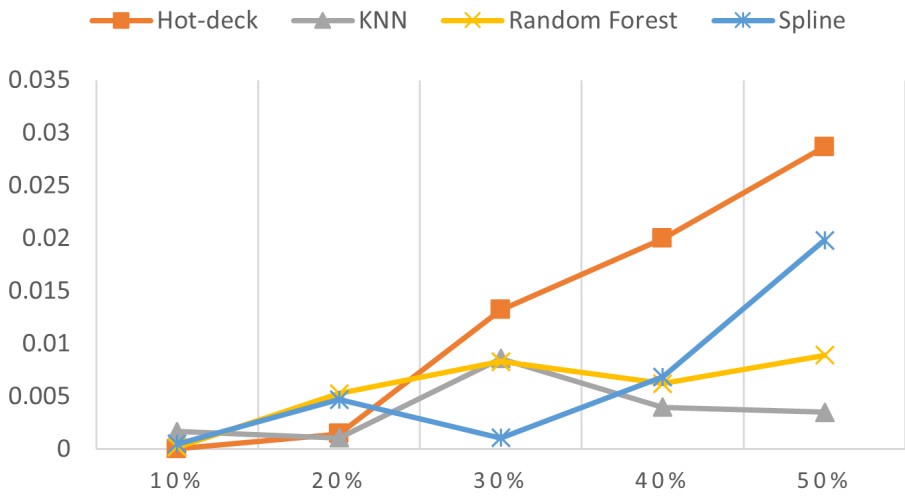

**Figure 10** MPEs of 5,882 records of daily gold price.

The MAEs of 5,882 records of daily gold price are presented in Fig. 8. KNN, random forest, hot deck, and spline imputation methods were compared. The results indicated that when the missing data percentage was 10%–50%, KNN had the smallest MAE among all the methods, followed by random forest. The spline method exhibited the worst performance among the four methods. Moreover, MAE increased when the missing data percentage increased.

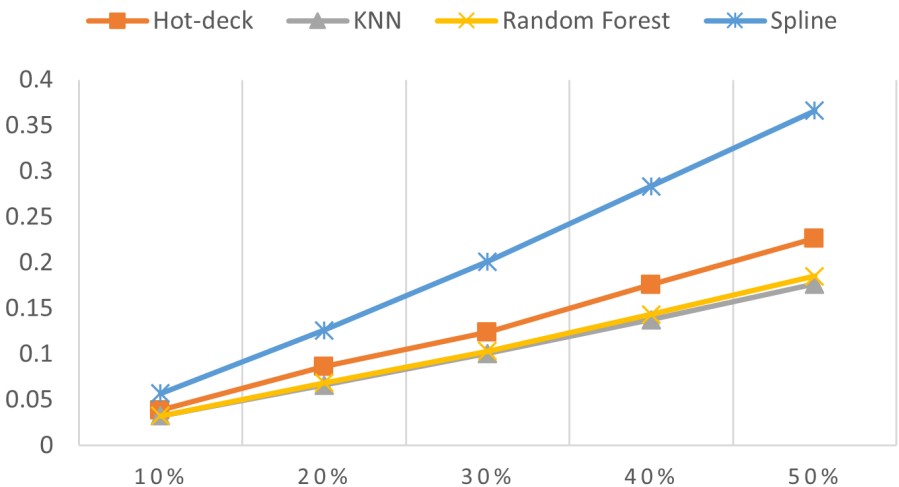

**MEAN ABSOLUTE PERCENTAGE ERROR**

**Figure 11** MAPEs of 5,882 records of daily gold price.

The RMSEs of 5,882 records of daily gold price are presented in Fig. 9. KNN, random forest, hot deck, and spline imputation methods were compared. The results revealed that when the missing data percentage was 10%–50%, KNN had the smallest RMSE, followed by random forest, among the four compared methods. The spline method demonstrated the worst performance among the methods. In addition, RMSE increased when the missing data percentage increased.

The MPEs of 5,882 records of daily gold price are given in Fig. 10. KNN, random forest, hot deck, and spline imputation methods were compared. As indicated in the figure, when the missing data percentage was 40%–50%, the KNN method had the smallest MPE, followed by random forest, among the four compared approaches. The spline method exhibited the best performance among the methods when the missing data percentage was 10%–30%. Moreover, an increase in the percentage of missing data resulted in an increment in MPE.

The MAPEs of 5,882 records of daily gold price are shown in Fig. 11. KNN, random forest, hot deck, and spline imputation methods were compared. The results revealed that when the missing data percentage was 10%–50%, KNN had the smallest MAPE, followed by random forest, among the four compared approaches. The spline method performed the worst among the four methods. When the percentage of missing data was increased, MAPE increased.

ME, MAE, RMSE, MPE, and MAPE were statistical measures that were utilized in order to assess the effectiveness of the various imputation techniques. By doing such a thorough examination, we were able to gain an understanding of the advantages and disadvantages of each strategy under a variety of different scenarios involving missing data percentages.

**Mean imputation**: This simple and computationally efficient method performed worst among all examined methods. Due to its variance reduction, this technique often biases

estimates, as shown by the greatest ME, MAE, RMSE, MPE, and MAPE across all missing data levels. Mean imputation fails to maintain data integrity under high missingness, as the fraction of incomplete data increases.

**Hot-deck imputation**: At lower missing data levels, hot-deck outperformed mean imputation. Although consistent, its performance declined as missing data increased. The method's reduced mistakes than mean imputation are due to its use of randomly selected neighboring values to preserve the dataset's distribution. Hot-deck imputation can still add bias, especially when huge datasets are missing.

**KNN imputation**: was the most reliable approach across missing data percentages. It continuously has the lowest ME, MAE, RMSE, MPE, and MAPE, especially with significant missing data percentages. The method uses nearest neighbors' values to impute values as close to the genuine values as possible, respecting data structure and distribution. This robust technique is ideal for datasets with changing missingness.

**Random forest imputation**: It also performed well, often matching KNN in accuracy. Ensemble learning, which aggregates several decision trees, handles complicated data relationships, reducing error rates. It performs marginally worse than KNN, especially in cases with more missing data. Non-linear variable correlations make random forest imputation beneficial.

**Spline imputation**: Spline interpolation worked moderately well, especially at lower missing data levels. As missing data accumulated, its performance declined. Fitting piecewise polynomials between data points with spline approaches can reveal data trends. They may have more errors than KNN and random forest imputation algorithms due to significant data gaps.

**SVM imputation**: SVM imputation performed inconsistently, generally ranking lower in accuracy. SVM is good at managing high-dimensional data and complex relationships, although imputation is difficult. Since it has larger error rates than other approaches, the method requires careful parameter calibration and may not be the ideal choice for imputation tasks.

**Missing data percentage impact**: The investigation shows that all imputation approaches are affected by missing data. Imputing huge amounts of missing data accurately is difficult since mistake rates rise with missing data percentage. The most resilient approaches to growing missing data percentages were KNN and random forest, with lower error rates.

**Financial time-series data implications**: The findings impact financial time-series data analysis. Correct missing value imputation is essential for accurate forecasting and decision-making. The superior performance of KNN and random forest approaches implies they should be used for financial dataset imputation. Their capacity to manage large missing data rates without degrading performance enables strong and trustworthy analyses.

## CONCLUSION

Missing data imputation was investigated using KNN, spline, hot deck, mean, and random forest imputation methods in this study. Five evaluation criteria, namely, ME, MAE,

RMSE, MPE, and MAPE, were employed to compare the methods' results. The KNN method demonstrated excellent performance and had a small error and high accuracy. Therefore, this method can be compared with any newly proposed imputation method.

As a result of the advantages that each approach possesses over the other, the new Imputation methods might be hybrid techniques that combine the methods that were employed in this research. Additionally, these hybrid methods might produce superior results. When it comes to predicting the future values of gold and currency prices, imputation methods can also be utilized as methods to make predictions.

## FUTURE WORK

Exploring datasets: Study how well the KNN imputation method performs on a variety of datasets, with missing data from fields like healthcare, finance and environmental science. This will help confirm its effectiveness across data types and characteristics.

Optimizing parameters: Examine how different parameters, such as the number of neighbors in KNN influence the methods performance. Conduct experiments to tune these parameters for improved accuracy and efficiency in the imputation process.

Utilizing methods: Look into the advantages of using methods that combine various imputation techniques, including KNN to capitalize on each methods strengths and enhance overall imputation accuracy.

Studying robustness: Evaluate how robust the KNN imputation method is under varying conditions like levels of missing data, presence of outliers or diverse data distributions. This analysis can offer insights into the methods reliability and stability across scenarios.

Real world implementation: Implement the KNN imputation method on real world datasets containing missing values and assess its performance in applications. Collaborate with experts in fields to gauge its utility in decision making processes and its impact, on analysis tasks.

Evaluate the KNN imputation method by comparing it to techniques, both traditional and machine learning based, on standard datasets. This comparison can reveal where KNN performs well and areas where it can be enhanced further.

Future research should prioritize investigating hybrid methodologies that integrate the advantages of several imputation techniques. Integrating KNN with random forest or spline interpolation has the potential to improve imputation accuracy. Furthermore, the enhancement of adaptive imputation techniques that can flexibly adapt to the specific attributes of the dataset and the type of missing data could potentially enhance performance even further.

Further investigation is needed to explore the potential use of these imputation techniques on many categories of financial data, including stock prices, currency rates, and commodity prices. Examining the effectiveness of imputation algorithms on different financial datasets will offer a more comprehensive understanding of their suitability and resilience.

### Funding

The authors received no funding for this work.

### Competing Interests

The authors declare there are no competing interests.

### Author Contributions

- Ala Alrawajfi conceived and designed the experiments, performed the experiments, analyzed the data, authored or reviewed drafts of the article, and approved the final draft.
- Mohd Tahir Ismail conceived and designed the experiments, performed the experiments, analyzed the data, authored or reviewed drafts of the article, and approved the final draft.
- Sadam Al Wadi conceived and designed the experiments, performed the experiments, authored or reviewed drafts of the article, and approved the final draft.
- Saleh Atiewi conceived and designed the experiments, performed the experiments, analyzed the data, performed the computation work, prepared figures and/or tables, authored or reviewed drafts of the article, and approved the final draft.
- Ahmad Awajan conceived and designed the experiments, prepared figures and/or tables, and approved the final draft.

### Data Availability

The Yahoo Finance (Gold Historical Price) (GC=F) dataset for Aug 30, 2000 to Feb 07 2024 is available at: https://finance.yahoo.com/quote/GC%3DF/history/?period1= 967608000&period2=1707264000.

### Supplemental Information

Supplemental information for this article can be found online at http://dx.doi.org/10.7717/ peerj-cs.2337#supplemental-information.

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
