# Peer review of "Multiple imputation methods: a case study of daily gold price"

_PeerJ Computer Science, doi:10.7717/peerj-cs.2337_

## Round 0.1 · original submission · Major Revisions

Based on the reviewers' feedback, we have decided that your paper requires a major revision before it can be considered for publication.

The reviewers have noted that certain phases of your study need to be clarified. Specifically, it is essential to provide more detailed explanations and justifications for these phases to enhance the overall comprehension of your research methodology and findings. Additionally, it is crucial to improve the experimental evaluation in accordance with the reviewers' comments. Please address each of their suggestions thoroughly to ensure that your study meets the high standards expected by our journal.

Moreover, we recommend that you carefully proofread your manuscript to resolve any typographical errors and other mistakes. A meticulous review of your paper will contribute significantly to its clarity and professionalism.

·

Basic reporting

The authors compared existing imputation methods in this study but did not invent any new methods. The authors need to add more details in almost all sections. Beginning with the abstract what the problems found in the daily gold prize and what the benefits gained by the current study are.

Add more reviews related to the imputation methods. What they did and what the results are in the introduction. Include some recommended works are as shown at the end of the comments.

Experimental design

In the real data set section add the citation and report how many percent of the missing values was found.

In section 5, what is the type of missing mechanism in this study?

In section 6, add more details for each method e.g. the models for each imputation technique.

Validity of the findings

Add more discussions in section 8. The results from tables only need a few digits shown.

Add more details in section 9 conclusion.

Additional comments

Consider to add more recent works in section 1 introduction as below.

K Chodjuntug and N Lawson (2022). Imputation for
estimating the population mean in the presence of nonresponse, with application to fine
particle density in Bangkok, Mathematical Population Studies, 29(4): 204-225.

K Chodjuntug and N Lawson (2022). A chain regression
exponential type imputation method for mean estimation in the presence of missing data,
Songklanakarin Journal of Science and Technology, 44 (4): 1109–1118

N Lawson (2023). New imputation method for estimating
population mean in the presence of missing data. Lobachevskii Journal of Mathematics,
44(9), 3740-3748.

N Lawson (2023). A class of population mean estimators
in the presence of missing data with applications to air pollution in Chiang Mai, Thailand.
Lobachevskii Journal of Mathematics, 44(9), 3749-3757.

N Thongsak and N Lawson (2023). A new
imputation method for population mean in the presence of missing data based on a
transformed variable with applications to air pollution data in Chiang Mai, Thailand,
Journal of Air Pollution and Health, 8(3): 285-298.

Cite this review as

·

Basic reporting

Mostly clear , unambiguous professional English has been used to write the case study. The abstract lays out the problem statement very well. The Introduction lays out the research topic in a very practical manner. However , before going into the research topic, It is important to establish what research
has already been conducted in this space through a literature review exercise. Although it is not completely absent , as lines 50 to 55 in the introduction touches upon it, it could have been expanded to illustrate what has been done in the past to address missing values and how this research will aid in what is already existing. Other than that , the article aligns with PeerJ standards,
The figures are labelled with legends and data files provided. However Figure 5 looks distorted and the readability is impacted. Figure 5 has been created using the Orange software. However 169 to 175 fails to provide a clear explanation of the Figure , why it was created , and what is the significance. There is one consistent feedback that needs to be rectified, the authors have used Abbreviations without providing the context. For example, MI in line 63 has no prior context. The authors have made an implicit
assumption that the reader is aware of the of the abbreviations , but that is open to interpretation and is
not recommended.

Experimental design

Overall experimental findings were well documented. In Line 102 - please explain “Infiling Strategies”, also reference to “Error! Reference source not 103 found” Is not very clear. Please rephrase correctly.

The mathematical model is very well constructed. With progressively injecting missing values from 10% to ultimately 50%. Also the charts show how the efficiency of each of those parameters (ME, MAE, RMSE, MPE) varies with the time-series data.Then , it would be nice to call where they deviate and why.

Validity of the findings

Under the discussion and results section, the observations of the experiment has been presented. It would be nice to discuss the significance of the findings, for example you have used ME or mean error for determining . In line 388, authors have mentioned that at high level imputations , KNN had performed relatively best and similarly at lower imputation levels , Spline Imputation performed the best. Now should it be considered satisfactory evidence to conclude that at higher missing frequencies , KNN is the best determinant of missing numbers in a time series model or will there be other factors such as MAR and RMAE that have a more explanation power.

Conclusion is very brief and misses a discussion on the relative performance of the 6 methods. What are the assumptions that have been made , going into the observations ? What considerations need to be made to apply this research across other time series missing imputations?

Additional comments

Overall it is a good article , with a strong area of research. The underlying business problem is very practical and has sound statistical considerations. Some of the review comments need to be taken into consideration to make this research at par with the industry standard.

Cite this review as

·

Basic reporting

Clear reporting. Figures, tables and raw data shared. Please refer "Additional comments" for more details.

Experimental design

Formal research question need to be provided so readers get the context of the study. The methods are described in detail, including data collection, the generation of missing values, and the evaluation criteria for imputation methods. Please refer "Additional comments" for more details.

Validity of the findings

The study compares existing imputation methods rather than proposing a novel technique. Results are presented statistically using multiple error metrics. Conclusion needs to be more specific. Please refer "Additional comments" for more details.

Additional comments

1. (Abstract) The abstract could be improved using a more structured format (e.g., Background, Methods, Results, Conclusion). At present, it lacks a clear statement of the study's objective and doesn't provide specific numerical results to support the claim.
2. (Introduction) The introduction section could benefit from a more logical flow. You may need to be reorganizing the start to introduce the broader context of gold prices and data analysis, then narrow down to the specific problem of missing data, and finally state the research objectives more explicitly.
3. (Introduction) The introduction should more clearly state the problem being addressed and the significance of this study. Is there a strong rationale for why missing data imputation in daily gold prices is critical for economic analysis and decision-making?
4. (Literature Review) The article does not include a dedicated literature review section. Can you include more recent literature reviews for imputation methods?
5. (Methodology) The cross-validation approach mentioned in Section 4 is not adequately detailed. You may need to provide more information about the specific cross-validation technique you used and how it was implemented to improve the robustness.
6. (Methodology) Line 102 – Fix this – “(Table 1)Error! Reference source not found.”
7. (Methodology) Can you provide detailed descriptions of the imputation methods used, including the specific parameters and settings for each method.
8. (Methodology) For dataset – can you provide description of the dataset ex. its source, the time covered, and if any data preprocessing steps taken before imputation. This is important to validate reproducibility of this study. Is this dataset publicly available? How did you access this data?
9. (Methodology) You may need to provide rationale for specific percentages of missing data (10%, 20%, 30%, 40%, 50%). Provide justification for these choices and discuss their relevance to real-world scenarios to determine feasibility of practical implications.
10. Figure 5- hard to read because of small size and figure 6 to 10 lack clear labelling.
11. (Evaluation Criteria) I can see evaluation metrics. Can you add more context why these metrics are suitable for assessing imputation performance?
12. (Discussion and Results) Can you add more analysis in Results section – ex. potential reasons for the observed performance differences between imputation methods. You may need to discuss the existing studies implications for the field of missing data imputation in financial time series analysis.
13. (Discussion and Results) Can you discuss the strengths and weaknesses of each imputation method in detail?
14. (Discussion and Results) Can you include computational efficiency of different imputation methods ex. processing time and resource requirements etc.
15. (Conclusion) In Conclusion – you may need to summarize the key findings and their significance. Include numerical results.
16. (Future Work) Section 10 covers some discussion; you may need to suggest specific future research directions and any potential improvements to the methodologies based on this study's limitations and results.
17. (Mathematical Model) Can you explain technical terms when they are introduced first. This would help readers who are not expert in specific imputation methods or statistical metrics.

Cite this review as

Reviewer 4 ·

Basic reporting

Some sentences and sections contain unclear English language. There are errors in certain references. Figures and tables are not well placed, making it not easy to link them to the text they explain. Additionally, there are formatting issues with the figures and tables.

There are more references on multiple imputation that should be considered. For example, I cannot see Rubin's rules or any reference to them in the article, despite "missingness mechanisms" are stated in section 5. The authors mentioned one recent reference but overlooked some of the most important ones such as:
Rubin and Schenker (1986): Rubin DB, Schenker N. Multiple imputation for interval estimation from simple random samples with ignorable nonresponse. J Am Stat Assoc. 1986; 81(394):366-74.
Rubin (1987): Rubin DB. Multiple imputation for nonresponse in surveys. 1987. New York: Wiley.

Additionally, it is unclear what raw data set was used and which variables are to be imputed using the suggested methods.

The results and conclusions are not properly summarised. The "future work" section is longer than the conclusion. The authors are invited to revise the findings and conclusion.

Experimental design

The abstract does not effectively highlight the key takeaways of the article.
The research question is not well defined, and there is a lack of clarity on how the research addresses knowledge gap in the multiple imputation framework. The research objective includes “identify trends,” but the article is addressing missing data imputation.

The background section fails to provide sufficient context on multiple imputation, and the literature review is inadequately referenced. The rationale for the selected methods is not well articulated, and there is no comparison with classic methods to validate multiple imputation.

The authors have not specified the missingness mechanism assumed in their research, despite defining these mechanisms theoretically in Section 5.

The methods are described with insufficient detail.

Validity of the findings

The data set section requires more clarification, particularly regarding whether a simulation was used. The authors are encouraged to better present the context of the “daily gold price” data.

It is unclear whether any simulations were conducted for this article.

I don't see how replicating the methods described in this article will be straightforward.

The results and conclusions are not properly summarised.

Additional comments

The article discusses the multiple imputation approach, which is used in a variety of domains, but it does not include several relevant references. While several imputation methods are used, the article does not explain the link between multiple imputation and the handling methods used. The multiple imputation methodology lacks a detailed description, including how the authors used it. Furthermore, the findings are insufficiently summarised to demonstrate how this article contributes to the knowledge gap in the application of multiple imputation in the financial market.

Cite this review as

---

## Round 0.2 · Minor Revisions

Thank you for submitting your manuscript to PeerJ Computer Science. The re-review process has now been completed, and we have received feedback from the reviewers.

I am pleased to inform you that, while the reviewers found your work to be of significant interest and quality, they have also identified a limited number of revisions that could further enhance the overall quality and clarity of the paper. These suggested revisions are minor but important for ensuring that your manuscript meets the highest standards of the journal.

We therefore invite you to submit a revised version of your manuscript that addresses the reviewers' comments. Please include a detailed response to each point raised by the reviewers, explaining how you have addressed their feedback or providing a justification if you have not made certain changes.

Once we receive your revised manuscript, we will proceed with the final stages of the review process.

·

Basic reporting

no comment

Experimental design

no comment

Validity of the findings

no comment

Cite this review as

·

Basic reporting

The review comments have addressed all pressing concerns. I have full confidence in accepting the paper . The main concern was in improving the figures to make it more details.

More context was demanded in terms of literature review. The corrections have addressed those concerns. The distortion in figure 5 has been corrected.
In methodology , the table names was inconsistent , which has now been fixed.
In the methodology section detailed steps were missing - they have been added and looks good now.

Experimental design

Overall , the sweeping changes made to the methodology section has now taken care of the detailed design of the experiment.

Validity of the findings

Thank you for incorporating all the changes. These changes have definitely improved the quality of the paper. Especially the Discussion and results section looks much better now .

Cite this review as

·

Basic reporting

This version looks good. I am accepting this article. Thanks for the fixes.

Experimental design

This version looks good. I am accepting this article. Thanks for the fixes.

Validity of the findings

This version looks good. I am accepting this article. Thanks for the fixes.

Additional comments

This version looks good. I am accepting this article. Thanks for the fixes.

Cite this review as

Reviewer 4 ·

Basic reporting

The authors have focused more on the mathematical scope of the various approaches. The article is of better quality with more context provided.

Certain sections and statements require some English revisions.

Certain graphs need to be formatted.

It was difficult to read the results with the graphs provided in the appendices.

Experimental design

The authors have provided more scope on the research methodology.
Could the authors supply the detailed summaries of the Yahoo Finance data from 2000-2024?

“The whole set of time-series data was subjected to a random simulation.” Which random simulation are the authors referring to here? Is it Gold data contamination with missing observations ?

Have the authors tested more than 1000 or less, for training datasets?

Validity of the findings

The findings are better clarified.

Abstract: Typo "because to its".

I would not claim that the KNN has "exceptional performance". In fact, the KNN results are better than the approaches used in this article, but they may not be as high performance as other methods.

Have the authors investigated the huge gaps between the accuracy measurements for SVM and Mean and Random Forest and KNN (referring to the results in Table 1 Accuracy Results for all Imputation Methods)?

Additional comments

Introduction: Unclear statement: "Authors investigate" from which article? is it (Chodjuntug and Lawson, 2022) ?

Research Objective: "Identify trends"; what trends were identified in this article; objective of this article is MI.

Which "stakeholders" are the authors referring to?

Please revise the statements:
“accurately and confidently navigate contemporary financial markets”
“This work applies a powerful and flexible imputation technique, namely, MI, to data imputation with a missing value estimation issue”

Figure 2: Can't this be inserted as the original flowchart rather than a snippet?

"Table 1: Sample data for the gold price (Yahoo, 2024)" but in the appendix "Table 1 Accuracy Results for all Imputation Methods". Please confirm.

kNN model/K-NN; please use the "KNN" or "K-NN" abbreviation consistently throughout the article.

Cite this review as

---

## Round 0.3 · accepted · Accept

Formatting: Please align the font formatting in section 6.6 SVM Imputation.

·

Basic reporting

Accept the changes , Article is now good for publication

Experimental design

Accept the changes , Article is now good for publication

Validity of the findings

Accept the changes , Article is now good for publication

Cite this review as

·

Basic reporting

Thanks for fixing comments. This version looks good, accepting this version.

Experimental design

Thanks for fixing comments. This version looks good, accepting this version.

Validity of the findings

Thanks for fixing comments. This version looks good, accepting this version.

Additional comments

Thanks for fixing comments. This version looks good, accepting this version.

Cite this review as

Reviewer 4 ·

Basic reporting

The authors provided the requested revisions.
Please include the x- and y-axis titles in Figure 1. What is the "P-Value test" for the Gold Price dataset? Which test is applied to this figure?
The recommendation for Table 1 was to include descriptive statistics.

Experimental design

No comments

Validity of the findings

No comments.

Additional comments

No comments.
Formatting: Please align the font formatting in section 6.6 SVM Imputation.

Cite this review as